# Is the Right to Housing Being Realized in Canada? Learning from the Experiences of Tenants in Affordable Housing Units in a Large Canadian City

Kaylee Ramage [1], Meaghan Bell [2], Lisa Zaretsky [3], Laura Lee [4] and Katrina Milaney [3,*]

[1] Community Health Sciences, Obstetrics and Gynecology, Cumming School of Medicine, Calgary, AB T2N 4Z6, Canada; kaylee.ramage@ucalgary.ca
[2] Calgary Housing Company, Calgary, AB T2G 2M1, Canada; meaghan.bell@calgary.ca
[3] Community Health Sciences, Cumming School of Medicine, Calgary, AB T2N 4Z6, Canada; lisa.zaretsky@ucalgary.ca
[4] Faculty of Social Work, University of Calgary, Calgary, AB T2N 1N4, Canada; laura.lee1@ucalgary.ca
[*] Correspondence: katrina.milaney@ucalgary.ca

**Abstract:** Background: Housing is a critical determinant of health and a basic human right. Historically, Canada's housing policies have not been grounded in a human rights-based approach. In the 1990s, a policy shift prioritized efficiency in government spending and deficit reductions over the provision of many social programs including affordable housing. With various levels of government now acknowledging and recognizing the need for more affordable housing, it is important to understand tenant experiences, perspectives, and needs to ensure policies and practices are supporting individuals appropriately. Methods: In total, 161 individuals participated in this study by completing online or in-person questionnaires. Results: Exploratory analysis of results revealed that although there were some positive benefits to affordable housing, many tenants continued to struggle financially, physically, mentally, and emotionally without adequate supports in place. Conclusions: These findings highlight the need for affordable housing to be part of a system of care that provides supports along a continuum. The results further reiterate that placing a person or family in affordable housing does not guarantee that their lives have improved. Without robust affordable housing models that prioritize the empowerment of individuals and families, housing policies may fail to fulfil the right to safe and affordable housing for Canadians, especially when considering historically marginalized populations.

**Keywords:** affordable housing; housing policy; human rights

## 1. Introduction

Housing is widely recognized as a necessity for health and wellbeing [1]. Housing is a significant determinant of health and a lack of housing has been associated with physical and mental health issues, high mortality rates, substance use disorders, limited access to primary healthcare, increased hospital visits, cycles of family poverty, and income insecurity [2–5]. Furthermore, the United Nations recognizes housing as a basic human right. Historically, Canada has failed to recognize its international human rights commitment to the right to safe and affordable housing, especially in relation to marginalized and vulnerable populations [6]. Recently, the Government of Canada has taken steps to progressively realize that housing rights are human rights through the establishment of the National Housing Strategy (NHS) which is governed by the National Housing Strategy Act (S.C. 2019, c. 29, s. 313). The NHS is a 10-year, CAD 40 billion plan focused on providing more than just a roof overhead by prioritizing investments in people, communities, and partnerships [7]. In 2018, the Government of Canada launched a national consultation to add a "human rights-based approach" to the original NHS. The NHS human

rights-based approach to housing is grounded in principles of accountability, participation, non-discrimination, and inclusion. Furthermore, the NHS human rights-based approach to housing includes the creation of an independent federal housing advocate within Canada's Human Rights Commission, a national housing council, community-based tenant initiatives, and a public engagement campaign. The purpose of these initiatives is to ensure people who have experienced homelessness and housing insecurity, along with allies or front-line workers in the field of housing programs or policies, are included in the NHS strategy [8]. As described in the national consultation, the absence of peoples lived experiences in the implementation of this strategy will ensure a failure to uphold a housing policy based on human rights [9]. In addition to providing the bricks and mortar, according to the National Housing Strategy Act, AF should protect "the inherent dignity and wellbeing of the person and to building sustainable and inclusive communities" (S.C. 2019, c. 29, s. 313).

The political and funding landscape for affordable housing in Canada has changed rapidly. Before the 1990s, the Canadian federal government maintained administrative and financial responsibility for housing in Canada. In the 1990s, there was an ideological and policy shift that prioritized efficiency in government spending and reducing deficits resulting in a disentanglement from many social programs including affordable housing [10]. Additionally, there was a lack of provincial will for public spending due to budgetary constraints [11]. Housing policy became increasingly uncoordinated, with provinces and municipalities creating their own policies in partnership with the private and community sectors [12]. As Pomeroy [13] indicates, these policies were not designed to be holistic or sustainable and resulted in subsidy dependency and suboptimal rent structures, uneven fiscal burden, a lack of a coordinated system, a disconnect between social service and business, and fragmented regulatory and governance frameworks. These shifts occurred simultaneously with a growing demand for affordable housing and increasing complexity in the health and social needs of those who required access to affordable housing options [14].

Throughout this paper we refer to social housing as affordable housing because social housing programs are included in the broader definition of housing affordability. Housing is considered affordable if it costs less than 30% of a household's pre-tax income [15]. Affordable housing is a broad term that can include housing provided by the private, public, and non-profit sectors. Furthermore, affordable housing may refer to housing on any part of the housing continuum including temporary shelters, transitional housing, supportive housing, subsidized housing, market rental housing, and market homeownership housing [16]. Affordable housing (AH) generally refers to programs where ongoing operating costs are paid by tenants and does not typically serve households with very low incomes [17]. Social housing is a form of affordable housing that is subsidized by the government to meet affordability requirements [15,18]. In Canada, social housing has traditionally been funded by multiple orders of government and a web of complicated agreements among different parties, including the private and not-for-profit sector. For further context, the Housing Services Corporation estimated that social housing totals 6% of Canada's housing market, compared to rates of 68% of home ownership and 26% of the private sector rental [17]. There is a disproportionate amount of market housing, which may or may not be affordable for different household incomes, in comparison to the lack of housing available to low-income Canadians, including in Calgary.

In 2016, the City of Calgary launched Foundations for Home, a 9-year corporate affordable housing strategy that prioritized the need for affordable housing [19]. The strategy recognized that housing is more than just a roof over one's head and focused on the importance of providing affordable housing that empowers individuals to achieve their goals, participate in the community, foster physical and mental wellbeing, and support them beyond housing. The 2018 Calgary Housing Needs Assessment [20] reveals almost 3000 people are homeless in Calgary with over 145,000 living in poverty. Furthermore, over 80,000 households struggle to pay for housing with this number expected to exceed 100,000 by 2025. Finally, the need for affordable housing is growing faster than what can be supplied, with key populations including Indigenous peoples, persons with disabilities,

lone-parent families, recent immigrants, seniors, singles, and youth. There is still much work to be done in addressing the need for affordable housing for low-income citizens.

There is currently limited evidence regarding specific tenant experiences, perspectives, needs, and trajectories through affordable housing. To address this knowledge gap and more specifically to help develop models of best practice in the design and delivery of affordable housing, this project sought to answer the following research questions: Who are the tenants of Calgary's Affordable Housing units? What are their health and service needs? What are the barriers and facilitators to moving from affordable housing into market housing? What is the relationship between affordable housing and homelessness?

## 2. Materials and Methods

Questionnaires were administered to tenants living in units operated by two of Calgary's largest AF developments to help understand lived experiences, trajectories through the housing continuum, and outcomes of affordable housing for families.

### 2.1. Participants

In total, 161 individuals participated in this study. Individuals were eligible to participate if they were over 18 years of age, had lived in an affordable housing unit for at least six months, and could respond to the questions in English. Posters were displayed at affordable housing sites and provided to property managers at the affordable housing units. Participants could complete the questionnaires online or in-person. Researchers visited multiple housing sites to provide the opportunity for in-person recruitment, attending community or on-site events. All potential participants were taken through the informed consent process (online or in-person), which informed them that their participation was voluntary and would not affect their service delivery, that their affordable housing provider would not know who had participated in the questionnaires, and that their results would be anonymized. Participants received a CAD 20 gift card to thank them for their time and participation. Ethics approvals were obtained through the University of Calgary Conjoint Health Ethics Review Board.

### 2.2. Materials

Questionnaires were designed in collaboration with a Community Advisory Committee (CAC) that included AH service providers, policy-makers, and researchers from housing initiatives in Alberta. The questionnaires included demographic information, housing trajectories, experiences in AH for the individual and their children, plans for their future, and gave participants an opportunity to provide feedback on their overall AF experience. This study was part of a collaborative community-based research partnership between the City of Calgary, Alberta Health Services (the provincial healthcare provider), and the University of Calgary, aimed at promoting health for Calgarians.

## 3. Results

Data was analyzed using Stata12. Variables were cleaned and coded based on self-reported information from the surveys. Qualitative responses (e.g., how would you define success?) were themed and recoded into numeric responses. Because of the small sample size, data analysis was explorative and descriptive, calculating number of responses and 95% confidence intervals for each question of interest. Table 1 presents the demographic characteristics of participants.

**Table 1.** Demographic characteristics of study participants.

| Variable | Categorization | Respondents (N) | Proportion (95% CI) |
|---|---|---|---|
| Gender | Female | 135 | 84.4% (77.8–89.2%) |
| | Male | 25 | 15.6% (10.8–22.2%) |
| Age | 18–24 years | 10 | 6.3% (3.4–11.4%) |
| | 25–34 years | 42 | 26.6% (20.2–34.1%) |
| | 35–49 years | 72 | 45.6% (37.9–53.4%) |
| | 50–65 years | 27 | 17.1% (12.0–23.8%) |
| | 65+ years | 7 | 4.4% (2.1–9.0%) |
| Ethnicity | Caucasian | 64 | 46.0% (37.9–54.4%) |
| | African or African Canadian | 30 | 10.8% (6.6–17.2%) |
| | Indigenous | 13 | 9.4% (5.5–15.5%) |
| | Asian | 13 | 21.2% (15.5–29.3%) |
| | Middle Eastern | 15 | 9.4% (5.5–15.5%) |
| | Other | N < 5 | Withheld |
| Marital Status | Single | 103 | 67.8% (59.9–74.8%) |
| | Partnered | 49 | 32.2% (25.2–40.1%) |
| Household Income | <$25,000 | 118 | 77.1% (69.8–83.1%) |
| | $25,001–$35,000 | 26 | 17.0% (11.8–23.9%) |
| | >$35,000 | 9 | 5.9% (3.1–11.0%) |
| Educational Attainment | Junior High School or Less | 12 | 8.3% (4.7–14.1%) |
| | Some High School | 30 | 20.7% (14.8–28.1%) |
| | High School Graduate | 37 | 25.5% (19.0–33.3%) |
| | Some Post-secondary | 31 | 21.4% (15.4–28.9%) |
| | Post-secondary Graduate | 35 | 24.1% (17.8–31.8%) |
| Have Children under the Age of Eighteen | Yes | 112 | 69.6% (62.0–76.2%) |
| | No | 49 | 30.4% (23.8–38.0%) |
| Born in Canada | Yes | 81 | 50.9% (43.2–58.7%) |
| | No | 78 | 49.1% (41.3–56.8%) |
| Immigration Status | Canadian Citizen | 45 | 60.0% (48.4–70.6%) |
| | Permanent Resident | 24 | 32.0% (22.3–43.5%) |
| | Temporary Resident or Refugee | 6 | 8.0% (3.6–16.9%) |
| Employed | Yes | 39 | 25.2% (18.9–32.6%) |
| | No | 116 | 74.8% (67.4–81.1%) |

Over two-thirds of respondents noted that the main reason they had moved into AH was because they could not afford the market housing price (64.6%, 95% CI: 56.9–71.6%). Overall, 37.3% (*n* = 57) of participants (95% CI: 29.9–45.2%) indicated that they were unable to work. Of those who reported being unable to work, 92.3% (95% CI: 72.5–98.2%) had a mental health concern in their household, 77.4% (95% CI: 63.9–86.8%) had a physical health issue in their household, and 57.4% had a disability in their household (95% CI: 42.7–71.0%).

Only 25.2% of the sample (95% CI: 18.9–32.6%) were employed, even though almost two-thirds reported being able to work. Of those who reported being able to work but being unemployed, 56.1% said they were unable to find a job (95% CI; 42.9–68.6%), 10.5% said they were ill (4.7–21.8%), 8.8% said they had a language barrier (95% CI: 3.6–19.7%), and 10.5% said they lacked appropriate transportation to get to work (4.7–21.8%). This lack of adequate employment required substantial reliance on financial assistance. Over one-third of participants received financial assistance through Alberta Works (35.4%, 95% CI: 28.4–43.1%) and 18.6% through AISH (95% CI: 13.3–25.5%).

Participants reported both positive and negative aspects of AH (Figure 1). Most tenants indicated that moving into AH led to less financial strain (51.5%, 95% CI: 43.8–59.2%). Moreover, 26.8% of respondents reported having more money since they moved into affordable housing (95% CI: 19.7–35.4%). Participants also reported positive impacts on their self-esteem, hope, feelings of safety, and education for their children. Approximately one-third of individuals said their employment had improved since moving into affordable housing. Furthermore, when looking at other non-housing outcomes of affordable housing, we found that approximately half of the tenants reported having improved quality of life, standard of living, and feelings of belonging. Conversely, 11.8% of the respondents (95% CI: 7.6–17.8%) reported that they were more stressed after moving into affordable housing. In addition, less than half of participants reported that their physical or mental health had improved since moving into affordable housing.

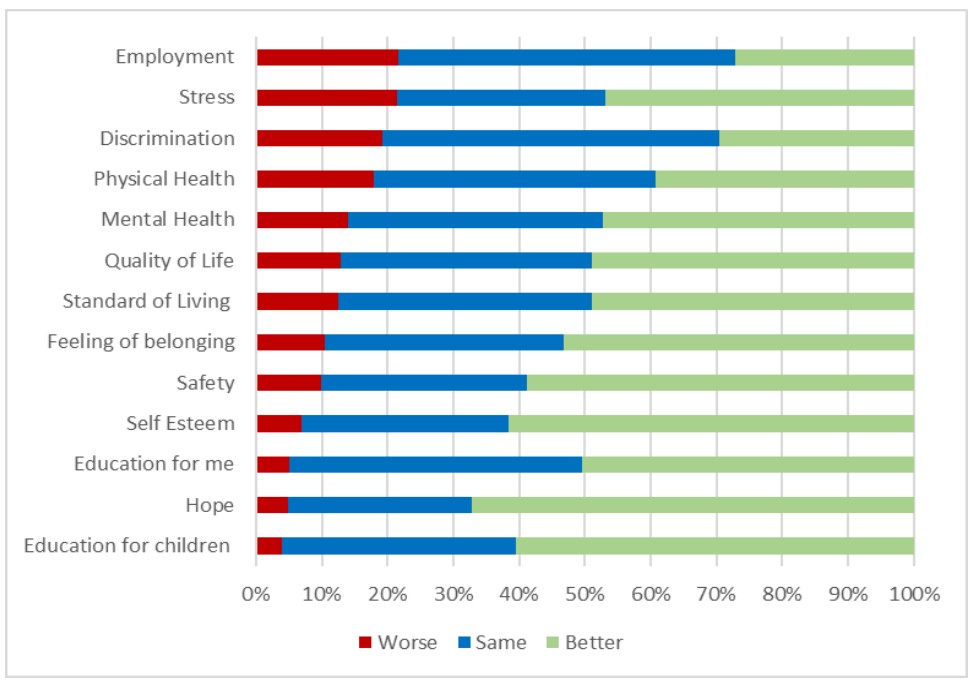

**Figure 1.** Tenant reported changes since moving into affordable housing.

Although this was a cross-sectional study, there was indication that affordable housing was an intergenerational experience, with many of the participants having family members who were also in affordable housing or who had lived in affordable housing as a child. Many participants reported having family members who were also in affordable housing or who had lived in affordable housing as a child. Of those respondents who were born in Canada, 19.7% had lived in affordable housing as a child (95% CI: 12.2–30.4%). As well, 19.1% of respondents overall said that they had a family member who was also living in affordable housing (95% CI: 13.4–26.6%). These intergenerational impacts also have the potential to affect the children who were living in affordable housing.

Finally, exploring tenants' housing history and outlook for the future indicated approximately one-third of participants had a lifetime history of homelessness (33.8%, 95% CI:

26.6–41.7%) and 16.8% (95% CI: 11.6–23.7%) of participants had moved directly from shelter into their current affordable housing unit. Others had moved from living with a family member (25.5%, 95% CI 19.1–33.2%), another affordable housing building (12.8%, 95% CI: 8.3–19.2%), or market housing (28.2%, 95% CI: 21.5–36.0%). Only 2.1% of participants said they expected to live in affordable housing for less than one year and only 14.1% of individuals said they expected to live in affordable housing for less than two years (95% CI: 9.2–20.9%). Most of the participants indicated that they would be living in affordable housing for more than five years (54.9%, 95% CI: 46.6–63.0%). Over three-quarters of the participants had lived in affordable housing for more than two years.

In relation to housing and financial stability, 55.3% of participants reported wanting to get a better job (95% CI: 47.5–62.8%) and 32.3% reported wanting to move into market housing (95% CI: 25.5–40.0%). However, most tenants who participated in our study focused on non-housing-related goals. When asked about their goals, almost half (42.9%, 95% CI: 35.4–50.7%) reported wanting their children to do well in school, 54.7% wanted to be less stressed about money (95% CI: 46.9–62.2%), and 42.9% wanted to get more education for themselves (35.4–50.7%). Over a quarter of participants (27.3%) reported that the rules governing their tenancy in their affordable housing unit affected their ability to reach their goals. When participants were asked what success looks like for them, 9.3% (95% CI: 5.7–14.9%) said better education, 17.4% (95% CI: 12.3–24.1%) said a better job, 5.6% (95% CI: 2.9–10.4%) said having a strong community connection, 21.1% (95% CI: 15.5–28.2%) said doing their best and trying hard, 24.8% (95% CI: 18.7–32.2%) said having a healthy family and children. Only 16.2% (95% CI: 11.2–22.7%) reported that success to them meant having more money.

## 4. Discussion

Our study on tenant experiences and expectations in AF in a large Canadian urban center provides important information to inform future service delivery and policy making for affordable housing delivery. Results suggest affordable housing in itself is not a solution for wellbeing or the realization of the human right to adequate housing. Although some tenants reported some positive benefits, many individuals were struggling and required more appropriate and well-rounded supports.

Some participants reported positive improvements in their self-esteem, hope, feelings of safety, and education for their children. Conversely, many participants continued to face significant difficulties which seemed almost insurmountable without additional supports. These difficulties included the inability to work due to mental health concerns, physical concerns, or a disability within the household. For those that were able to work, over half of participants said they could not find a job, encountered language barriers, and were unable to access transportation to get to work. Many participants also had a history of homelessness, with some moving from shelters into AF units. These results support the argument that AH must provide more than just a physical structure [19]. Additional supports should ensure individuals are connected to appropriate supports in the community for mental and physical wellbeing, access to career and employment guidance, language classes, and access to affordable transportation.

Similar to a recent study from the Australian Housing and Urban Research Institute (AHUR) [21], our results show that over half of tenants did not consider their social housing tenancy to be temporary, with many planning to live there for the foreseeable future. Although we lacked the sample size for complex modelling or examination of the patterns of entry and exit, our findings closely aligned with those identified in the AHUR study, which found that tenants who do move on, moved to spaces with less residential stability (rather than to the more stable home ownership).

With limited transitions out of housing, the growing and persistent need for housing, and few new AF housing units being added, waitlists for social housing continue to grow. Research on AH has tended to focus on housing policy analysis without substantial efforts to generate best practices in provision of social or affordable housing for operators

or tenants [22]. There is a lack of consensus on the intended outcomes of AH which has created challenges in the design and delivery of support programs. Opportunities to integrate evidence and ground service delivery in an intentional program around housing as a human right or part of a poverty reduction strategy have been overlooked. Housing has continued to take an approach of universality centered on provision of "brick and mortar" housing, rather than tailored interventions to acknowledge the diversity and intersectionality of experience and needs of tenants today. Examining Canada's approach to housing policy suggests the market failures contributing to the increased demand for affordable housing stems from growing income inequality, the loss of low-rent housing stock through gentrification, and the loss of land zoned specifically for rental housing [1]. While there is no doubt that more AH is needed, clarity on what affordable housing can or should deliver is also needed. Perhaps traditionally conceived of in the 1970s through federal/provincial operating agreements as "low-cost" housing for a relatively homogenous population, social housing now exists in a complex web of policies and serves an incredibly diverse population with multiple health and social needs. Clarifying the objectives for social and affordable housing and understanding the interplay between housing and other sectors such as homelessness, family violence, early childhood development and poverty reduction is necessary to achieve effective planning and policy responses.

Canada's National Housing Strategy calls for a human rights-based approach to housing that is grounded in principles of accountability, participation, non-discrimination, and inclusion [8]. If AH is intended to ensure the dignity and wellbeing of residents as well as build inclusive communities, it seems Calgary's approach has only been partially successful. Our evidence suggests that stable housing offered within a coordinated "system of care" is the first and most important step to healthy families and healthy communities and that without housing and supports to sustain it, interventions to reduce poverty, improve access to education, employment, and healthcare are ineffective and often impossible [23]. AF service providers should either embed healthcare policy and social services within their own program models or ensure streamlined access and strong partnerships with local health authorities and not for profits to ensure tenants have free or affordable access to holistic supports based on individual needs.

The results of this study indicate that more support and tailored interventions are needed to address the economic, social, and health inequities of AH tenants to support a successful flow-through from social housing into market housing [24]. Our sample provides an unsurprising overview of a profile of tenants, largely unemployed and on social assistance, mostly single, female-headed households with high rates of physical and mental health challenges and high rates of histories of precarious housing, often intergenerational. With regards to intergenerational disadvantage, research suggests unaffordable housing has implications on child and youth physical, mental, and social health and development. Furthermore, high housing costs have been associated with food insecurity and poor child nutrition [23]. Finally, the link between poverty and inadequate housing continues to be perpetuated when housing is dismissed as a public health issue [25]. Our findings show that while tenants in AF may have a roof over their heads, they are still living in poverty and often have healthcare issues that prevent them fully participating in society. Even with part-time or full-time employment in addition to AF, this is not a guarantee that their human rights have been actualized. Participants indicated future aspirations related to employment and moving out of social housing; thus, embedded human rights strategies within social housing programs are needed as the current system of only providing low-cost housing seems to only perpetuate the cycle of poverty, dependency, and ill health across generations. Furthermore, policy-makers must recognize the health implications associated with housing and ensure policies are developed to provide affordable housing for children, youth, and families [23]. COVID-19 has only exacerbated the inequities in housing and supports the urgency of ensuring all Canadians have the right to safe, adequate, and affordable housing [26].

There are several limitations to this study in addition to limitations with descriptive analysis. Our cross-sectional study only captures a point in time in tenant experiences, longitudinal research could provide deeper insights into pathways, trajectories, and changes over time. A relative lack of diversity among our participants limits what can be known about the intersectionality of poverty and health with culture, race, and multiple identities. Qualitative research that focusses on tenant experience in richer detail could also provide important insights into specific interventions that could be implemented and evaluated. Inclusion of AH staff and administrative bodies like municipalities or organizations that run AH programs could shed light on resource issues and/or specific policy change ideas.

## 5. Conclusions

Our study provides important evidence to support the need for housing policy change in Calgary, Canada and the need to redesign the AH system to better support the health, wellbeing and human rights of tenants. Utilizing tenants' perspectives and experiences, we are one of the first studies in Canada to examine this important issue, especially from the perspective of the tenants themselves. Tenants are not receiving adequate supports once they are housed in affordable housing and we must find ways to build capacity to support them. Housing providers and policy-makers need to be engaging with the NHS while also developing an evaluative strength-based framework to support people to improved health and wellbeing outcomes. Using this study as a starting point, future policy change or research should seek to implement and evaluate initiatives or housing models that prioritize an increase in overall wellbeing that tenants experience while living in affordable housing. Housing is more than a basic need, it is a human right. The lack of decent, safe, affordable, and integrated affordable housing is a significant barrier to participation in community life.

**Author Contributions:** Conceptualization, K.M., M.B. and K.R. methodology, K.M., K.R. and M.B.; formal analysis, K.R. and L.Z.; resources, K.M.; data curation, K.R.; writing—original draft preparation, K.M., K.R., M.B., L.Z., L.L.; writing—review and editing, K.M. and L.Z.; supervision, K.M. project administration, K.M. and K.R. funding acquisition, K.M. All authors have read and agreed to the published version of the manuscript.

**Funding:** This research was funded by MakeCalgary Initiative.

**Institutional Review Board Statement:** This study was approved by the University of Calgary Conjoint Health Research Ethics Board REB 17-0816.

**Informed Consent Statement:** Informed consent was obtained from all subjects involved in the study.

**Data Availability Statement:** Data is accessible through the corresponding author assuming eth-ical approvals have been obtained.

**Acknowledgments:** The authors wish to thank our funders and our Advisory Committee members who made this study possible. We would also like to thank to the participants in our study who shared their experiences.

**Conflicts of Interest:** The authors declare no conflict of interest. The funders had no role in the design of the study; in the collection, analyses, or interpretation of data; in the writing of the manuscript, or in the decision to publish the results.

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
