# Peer review of "Is the Right to Housing Being Realized in Canada? Learning from the Experiences of Tenants in Affordable Housing Units in a Large Canadian City"

_societies, doi:10.3390/soc11020053_

Round 1
Reviewer 1 Report
A good study about the limitations of Affordable/social housing in Canada, with a Calgary case study. The methodology is sound. The criticism of the current approach based on the responses of the study participants is sound. The limitations of the study are clear. If, in addition to the cited Australian study, there have been any similar comparative Canadian studies, it would be helpful to know if in other Canadian provinces a different approach like that advocated by the authors has be taken.
Author Response
As per your comment, "in addition to the cited Australian study, there have been any similar comparative Canadian studies, it would be helpful to know if in other Canadian provinces a different approach like that advocated by the authors has be taken."
Response: While there are likely provincial advocates arguing for a human rights based approach to AF - we were unable to find literature to support. We are not aware of any other studies in Canada that have looked at social housing pathways. We feel our study is a starting point for other jurisdictions to explore in future research.
Reviewer 2 Report
As the authors made the claim that a more extensive study is needed to examine the complexities of problems that people in public housing face. Housing cannot be separated from other aspects of living. Employment, health, social connections, are all essential.
One finding of the study that is interesting, as well as having policy implication, is the problem of intergenerational disadvantage. What should/could the government, as well as citizens do to address this issue? How can children from disadvantaged families, move beyond the cycle of social and economic exclusion?
This situation must have been made worse because of the pandemic, as well as the economic challenges that Alberta, and Calgarians are facing.
Author Response
We have added references that address the intergenerational disadvantage, and speak to the link between unaffordable housing and child/youth development and the current pandemic. The paragraph has been updated as follows:
With regards to intergenerational disadvantage, research suggests unaffordable housing has implications on child and youth physical, mental, and social health and development [40]. Furthermore, high housing costs have been associated with food insecurity and poor child nutrition [40]. Finally, the link between poverty and inadequate housing continues to be perpetuated when housing is dismissed as a public health issue [40]. Our findings show that while tenants in AF may have a roof over their heads, they are still living in poverty and often have health care issues that prevent them fully participating in society. Even with part-time or full-time employment and to AF, this is not a guarantee that their human rights have been actualized. Participants indicated future aspirations related to employment and moving out of social housing; thus, embedded human rights strategies within social housing programs are needed as the current system of only providing low-cost housing seems to only perpetuate the cycle of poverty, dependency, and ill health across generations. Furthermore, policy makers must recognize the health implications associated with housing and ensure policies are developed to provide affordable housing for children, youth, and families [40]. Covid-19 has only exacerbated the inequities in housing and supports the urgency of ensuring all Canadians have the right to safe, adequate, and affordable housing [41].
Reviewer 3 Report
The paper addresses tenants’ experiences, perspectives, needs and trajectories through affordable housing in Calgary. The theoretical perspective of the study is based on the human rights approach according to which housing is grounded in principles of accountability, participation, non-discrimination, and inclusion. In the introduction the author/s present the current situation in Canada in the regard to housing policies and changes in legislation. They also critically evaluate the gaps and deficits in the implementation of these polices.
The empirical section of the paper is based on a descriptive analysis of the results form a survey conducted with a small sample of tenants, who responded to questions regarding their experiences with of social housing. The design and the limitations of the study are clearly presented and discussed.
The discussion of the results clearly outlines the deficits in housing policy based on the experiences and the opinions of the respondents and draws on important conclusions for the deficits of housing policy. One of the main conclusions is that a holistic supports based on individual needs are needed in order to address poverty and break the circle of disadvantage. In my opinion the paper contributes to existing literature on social rights, poverty and housing that is limited in terms of empirical studies. It also inspires new insights for future research on this topic.
My suggestion to the author/s of the paper is to comment more explicitly the ‘positive’ and the ‘negative’ evaluations of the tenants in regard to social housing in the ‘Results’ part of the paper since at this version the opinions are mixed. Such regrouping would help the reader to understand better the opinions of the respondents on different aspects of life in the social houses.
Author Response
In the Results section, the following paragraph has been added:
Participants reported both positive and negative aspects of AH (Figure 1). Most tenants indicated that moving into AH led to less financial strain (51.5%, 95% CI: 43.8-59.2%). Moreover, 26.8% of respondents reported having more money since they moved into affordable housing (95% CI: 19.7-35.4%). Participants also reported positive impacts on their self-esteem, hope, feelings of safety, and education for their children. Approximately one-third of individuals said their employment had improved since moving into affordable housing. Furthermore, when looking at other non-housing outcomes of affordable housing, we found that approximately half of the tenants reported having improved quality of life, standard of living, and feelings of belonging. Conversely, 11.8% of the respondents (95% CI: 7.6-17.8%) reported that they were more stressed after moving into affordable housing. In addition, less than half of participants reported that their physical or mental health had improved since moving into affordable housing.